# Comparative Profiling of Hot and Cold Brew Coffee Flavor Using Chromatographic and Sensory Approaches

**DOI:** 10.3390/foods11192968

**Published:** 2022-09-22

**Authors:** Yanpei Cai, Zhenzhen Xu, Xin Pan, Min Gao, Mengting Wu, Jihong Wu, Fei Lao

**Affiliations:** 1National Engineering Research Center for Fruit and Vegetable Processing, Key Laboratory of Fruit and Vegetable Processing, Beijing Key Laboratory for Food Non-thermal Processing, Ministry of Agriculture and Rural Affairs, College of Food Science and Nutritional Engineering, China Agricultural University, Beijing 100083, China; 2Institute of Quality Standard & Testing Technology for Agro-Products, Chinese Academy of Agricultural Sciences, Key Laboratory of Agro-food Safety and Quality, Ministry of Agriculture and Rural Affairs, Beijing 100081, China

**Keywords:** coffee, aroma, non-volatile, chromatography, sensory profile

## Abstract

Coffee brewing is a complex process from roasted coffee bean to beverage, playing an important role in coffee flavor quality. In this study, the effects of hot and cold brewing on the flavor profile of coffee were comprehensively investigated on the basis of chromatographic and sensory approaches. By applying gas chromatography–mass spectrometry and odor activity value calculation, most pyrazines showed higher contribution to the aroma profile of cold brew coffee over hot brew coffee. Using liquid chromatography, 18 differential non-volatiles were identified, most of which possessed lower levels in cold brew coffee than hot brew coffee. The sensory evaluation found higher fruitiness and lower bitterness and astringent notes in cold brew coffee than hot brew coffee, which was attributed by linalool, furfural acetate, and quercetin-3-O-(6″-O-p-coumaroyl) galactoside. This work suggested coffee brewing significantly affected its flavor profile and sensory properties.

## 1. Introduction

Coffee (*Coffea* sp.) is the second most globally traded commodity after petroleum [1] and accounts for broad market and high economic values. It is estimated that the worldwide production of coffee increased to 9.9 million tons in 2019 [2]. Coffee is popular among consumers due to its pleasant flavor profile. The flavor quality of coffee is principally determined by chemical components including volatile compounds, which contribute to the aroma properties, and non-volatile compounds to the taste. These flavor compounds can be significantly affected by many complex factors, such as processing method, roasting degree, and brewing method, which all consequently affect the consumer choice [3].

In recent years, the brewing process, as the final step closest to coffee consumption, is increasingly attracting the interest of food manufacturers. Extraction parameters play key roles in coffee sensory quality and flavor perception, such as the ground coffee particle size, extraction time, pressure, and temperature [4]. On the basis of the brewing temperature [5], it can be divided into classical hot brew coffee and novel cold brew coffee (≤25 °C). As reported by most sensory studies, cold brew coffee has been considered as fruity, floral, and sweet [5,6], while hot brew coffee possesses signature roasted and smoky-like smells [7].

Generally, the high temperature increases the kinetic energy of water molecules, which favor the leaching out of chemical compounds from the coffee powders. Meanwhile, the high temperature can affect the release of volatile compounds in coffee, thereby altering the sensory perception [8]. With advanced analytical techniques, chromatography–mass spectrometry was applied to compare the differences of physiochemical compounds between hot and cold brew coffee. In a comparative study on the volatiles in cold and hot brew Arabica coffee from Colombia with the medium roasting degree [9], it was reported that cold brew coffee with immersion extraction had a higher abundance in total furans and pyrazines than hot brew coffee by the French press method. As for non-volatiles, our previous study reported that the contents of physiologically functional norhorman and harman were richer in hot brew coffee with pour-over and boiled methods than cold brew coffee using liquid chromatography quadrupole time-of-flight mass spectrometry [10]. Interestingly, the hot brew coffee exhibited significantly higher bitterness and astringency over cold brew coffee [11], but no significant difference was detected in the quantities of a few common compounds associated with coffee bitterness, such as total caffeoylquinic acids, caffeine, and trigonelline [11,12]. Yet, a significant difference was frequently reported in previous sensory studies [4,5], especially in the flavor perception, between hot and cold brew coffee. Until now, very limited knowledge is available on the compounds that are responsible for the differential sensory profile between hot and cold brew coffee. Moreover, the coffee preparation in most studies referred to homemade process without standardization after coffee extraction, which would be of little help in understanding the coffee flavor for industrial processing. Therefore, a comprehensive analysis is needed.

The objective of this study was to compare the aroma-active compounds in hot and cold brew coffee by gas chromatography–mass spectrometry (GC-MS) and odor activity value calculation, to differentiate non-volatile compounds including organic acid, sugar, and phenolic compounds using the liquid chromatography with pulsed amperometric or mass detector and multivariate statistical analysis. Lastly, the sensory triangle test and quantitative descriptive analysis (QDA) of hot and cold brew coffee were conducted to tentatively figure out potential correlations between differential flavor compounds and sensory attributes of hot and cold brew coffee prepared on the basis of industrial processing. 

## 2. Materials and Methods

### 2.1. Materials and Chemicals 

The same commercial batch of medium-dark roasted Arabica coffee beans (Pacific, Shanghai, China) blended from Costa Rica, Colombia, and Indonesia were selected for coffee brewing, which were purchased from the local market in January 2020 in Beijing city. A total of 2.0 kg of coffee beans were fully mixed right before coffee preparation.

High-performance liquid chromatography (HPLC) grade of acetonitrile and methanol were purchased from Thermo-Fisher (Waltham, MA, USA). Formic acid, quinic acid, shikimic acid, malic acid, citric acid, tartaric acid, and oxalic acid were of HPLC grade and purchased from Sigma-Aldrich (St. Louis, MO, USA). HPLC grade of glucose, sucrose, fructose, maltose, arabinose, and galactose were from Sinopharm Chemical Reagent Co., Ltd. (Beijing, China). The 3-octanone standard was from TCI Chemical Industry (Tokyo, Japan). Other reagents used were of analytical grade and bought from Sinopharm Chemical Reagent Co., Ltd. (Beijing, China). 

### 2.2. Coffee Preparation

Coffee beans were grounded by the semi-automatic grinder (E10, HERO, Beijing, China) and sieved to obtain 30-mesh to 60-mesh coffee powders for hot and cold brew coffee preparation. On the basis of previous studies with some modifications [13,14], a coffee machine (NC-F400, Panasonic, Tokyo, Japan) was used for hot brewing with a water-to-powder ratio of 10:1 (*w*/*w*). Briefly, 60 g of coffee powders were placed on a coffee filter paper (No. 4, Melitta, Germany) after 600 g of purified water was added into the machine. The water was heated to brew coffee with the outlet coffee extract temperature at 80 ± 3 °C, with the brewing time of approximately 6 min. For cold brew coffee, 60 g of coffee powders were placed into a 300-mesh filter bag and let soak in 600 g of purified drinkable water in a glass jar at room temperature (20 ± 3 °C) for 8 h, with the jar lid closed. Additional 120 g purified drinkable water was applied to rinse coffee powders and the rinsed liquid was collected into the jar and mixed with previous obtained cold brew coffee. The liquid in the jar was filtered through a coffee filter paper before sending for stock. Referring to industrial processing, hot and cold brew coffee were diluted to standardize to 2% total dissolved solids (TDS) for sensory and subsequential instrumental analysis, which was consistent with previous studies [6,12]. Hot and cold brew coffee preparation was performed in three replicates for each independent experiment.

### 2.3. Volatile Compound Analysis 

#### 2.3.1. Extraction of Volatile Compounds

Hot and cold brew coffee samples were sent to GC-MS to determine the volatile compounds. The extraction of volatile compounds in coffee was conducted using solid phase microextraction (SPME) as described by Pan et al. [15], with slight modifications. Coffee (4 mL) was transferred into a 20 mL headspace bottle containing 2.5 g NaCl and 50 μL of 3-octanone (8.2 mg/L in methanol, as internal standard). The bottle was sealed by PTFE-silicone septum and equilibrated at 60 °C for 10 min with agitation. Next, a 50/30 μm divinylbenzene/carboxenTM/polydimethylsiloxane SPME fiber was exposed to the headspace of the coffee for 45 min at the same temperature without stirring. Finally, the fiber was obtained and introduced into the gas chromatography injector at 250 °C for 5 min.

#### 2.3.2. Gas Chromatography–Mass Spectrometry 

GC-MS analysis was carried out according to the method described by Heo et al. [14] using an Agilent 7890 gas chromatography system (Agilent Technologies, Santa Clara, CA, USA) equipped with an Agilent 5975C series mass spectrometer. The volatile compounds were isolated with HP-INNOWAX (30 m × 0.25 mm i.d. × 0.25 μm; Agilent Technologies) fused silica capillary columns. The carrier gas was helium at a rate of 1 mL/min constant flow. The oven temperature was held at 40 °C for 10 min, ramped at the rate of 8 °C/min to 180 °C, followed by ramping to 280 °C at the rate of 10 °C/min for 10 min. Mass spectrometry was carried out at an electron impact mode of 70 eV with a scan range of m/z 33–550. The volatile compounds were identified on the basis of the matching of the mass spectra with those in the standard NIST10 database. The quantification of volatile compounds was performed using peak areas normalized with 3-octanone added to each sample as an internal standard. 

#### 2.3.3. Identification of Aroma-Active Compounds

Odor activity value (OAV) was the ratio of concentration to odor threshold [16]. In most cases, the OAV value > 1 means the compound contributes to the overall coffee aroma profile. 

### 2.4. Non-Volatile Compound Measurement

#### 2.4.1. Organic Acid and Sugar Profile Analysis

Coffee samples of 5 mL were centrifuged at 9000× *g* for 5 min, and the supernatant was collected for chromatographic analysis. According to previous reports [17,18] with some modification, the supernatant was passed through a 0.22 μm filter membrane before being injected into a Diane ICS-3000 ion chromatography coupled with a pulsed amperometric detector for analysis.

Acid chromatographic separation was achieved using the ion chromatography column (Dionex IonPacTM AS11-HC, 4 × 250 mm), with 20 μL injection volume and 0.60 mL/min flow rate, at 65 °C. Sulfuric acid solution (0.005 mol/L) was set to the eluent. External Standard solutions were applied for qualitative and quantitative analysis, which were quinic acid, shikimic acid, malic acid, tartaric acid, oxalic acid, and citric acid. 

Sugar chromatographic separation was conducted by the ion chromatography column (Dionex CarboPacTM PA20, 3 × 150 mm, pre-PA20, 3 × 30 mm guard column), with the 30 °C column temperature. The solvent system was water (eluent A) and 200 mmol/L NaOH solution (eluent B): 0–20 min, 95% A; 20–30 min, 80% A; 30–40 min, 0% A; 40–50 min, 95% A. The flow rate was 0.4 mL/min, and the injection volume was 20 μL. Arabinose, galactose, glucose, sucrose, fructose, and maltose external standard solutions were also applied. Retention times of the external standard solution were used for qualitative analysis, and peak areas were used for quantitative analysis.

#### 2.4.2. Ultra-Performance Liquid Chromatography–Tandem Mass Spectrometry (UPLC-MS/MS)

The extract was vortexed for 5 min, 500 μL coffee was added to 500 μL of 70% methanol solution, and then the extract was vortexed for 15 min. Following centrifugation at 12,000× *g* for 3 min, the extract was filtered through a 0.22 μm filter (SCAA-104; ANPEL, Shanghai, China) before analysis by UPLC-MS/MS.

According to Li et al. [19], with slight modification, non-volatile analysis of hot and cold brew coffee was carried out on a UPLC-MS/MS system consisting of a Nexera X2 UPLC system (Shimadzu, Kyoto, Japan), connected to a 4500 QTRAP mass system (Applied Biosystems, Waltham, MA) equipped with an electrospray ion source (ESI). The SB-C18 column (2.1 × 100 mm, 1.8 μm; Agilent Technologies, Santa Clara, CA, USA) was used for the chromatographic separation at 40 °C. The binary mobile phase consisted of water with 0.1% formic acid (eluent A) and acetonitrile with 0.1% formic acid (eluent B). The flow rate was 0.35 mL/min, and the gradient elution was carried out as follows: 0–9.00 min, 5−95% B; 9.00–10.00 min, 95% B; 10.00–11.10 min, 95−5% B; 11.10–14.00 min, 5% B. Samples (4 μL) were eluted at 0.45 mL/min.

Linear ion trap and triple quadrupole scans were performed using a triple quadrupole–linear ion trap (QTRAP) mass spectrometer (AB4500 QTRAP UPLC/MS/MS System). Sequence analyses were carried out in positive ion mode (ion spray voltage of 5500 V) and negative ion mode (ion spray voltage of −4500 V) with an ion source temperature of 550 °C. The pressures of the ion source gases I and II and the curtain gas were set at 345, 414, and 172 kPa, respectively. Instrument tuning and mass calibration were performed with 10 and 100 μmol/L polypropylene glycol solutions in triple quadrupole and linear ion trap modes, respectively. The data were collected at a scan range of m/z 100−1200.

The triple quadrupole scans were performed as multiple reaction monitoring (MRM) mode with the pressure of the collision gas (nitrogen) set to medium. The declustering potential and collision energy were selected for the individual MRM transitions and were then further optimized. A specific set of MRM transitions were monitored for each period according to the compounds eluted within this period. 

### 2.5. Color Analysis

Colors of the hot and cold brew coffee were measured using a spectrophotometer (Hunterlab ColorQuest XE) at the transmission mode, D65 illuminant, 0.375 inch observation aperture, and 10° observation angle. A cuvette with a light path of 5 mm was applied in this study. The L*a*b* values of hot and cold brew coffee samples were determined after calibration. L* represents lightness, and a* and b* are chromaticity indexes. The total color difference (ΔE) of the two coffee was calculated via
$$\Delta E=\sqrt{{\left({L^{*}}_{\mathrm{cold}}-\;{L^{*}}_{\mathrm{hot}}\right)}^{2}+{\left({a^{*}}_{\mathrm{cold}}-{a^{*}}_{\mathrm{hot}}\right)}^{2}+{\left({b^{*}}_{\mathrm{cold}}-{b^{*}}_{\mathrm{hot}}\right)}^{2}}$$

### 2.6. Sensory Evaluation

All sensory evaluation was conducted at room temperature. Coffee samples were brought out from the refrigerator and let set at room temperature until 20 ± 3 °C for panel tasting.

#### 2.6.1. Sensory Triangle Test

A total of 48 university students were recruited (age 19–22, 18 males and 30 females) for the sensory triangle test. According to ISO [20], three randomly coded hot and cold brew coffee samples were presented to the sensory panelist, two of which were the same and the other was different. The sensory panel was asked to select the sample different from the others. The test was carried out under red illuminant in order to avoid the interference of the coffee color difference. 

#### 2.6.2. Sensory Quantitative Descriptive Analysis

The descriptive sensory panel consisted of 7 university students (age 19–22, 2 males and 5 females) with regular coffee consumption at least 3 years. All panelists previously participated in at least 3 rounds of training on the identification and differentiation of the basic sensory attributes in coffee. To train panelists to understand sensory attributes of coffee, description vocabulary and their intensity scoring standards of coffee sensory evaluation were as shown in Appendix A [5,21]. After training, intensities of sensory attributes in hot and cold brew coffee were evaluated by the panel. Consistent with other sensory quantitative descriptive analysis [22], the panel was asked to evaluate coffee visual appearance and gave a score for color under the natural lighting condition, then took a deep smell of the coffee to evaluate the intensity of aroma before drinking it to evaluate its taste. The sensory description vocabulary of coffee included dark-colored, nutty aroma, coffee aroma, fruity aroma, sweetness, sourness, bitterness, and astringency, with the intensity score ranging 1–5 points.

### 2.7. Data Processing and Multivariate Statistical Analysis

Data were reported as mean ± standard deviation. Student’s t-tests were performed using SPSS Statistics 26 (IBM, Armonk, NY, USA) at a 95% significant level.

The LC-MS/MS raw data signals were processed by Analyst 1.6.3 software (AB Sciex, Framingham, MA, USA). Principal component analysis (PCA), hierarchical cluster analysis (HCA), and orthogonal partial least-squares discriminant analysis (OPLS-DA) were carried out by the R software package (www.r-project.org) to visualize the differences between hot and cold brew coffee. Heatmap analysis and cluster analysis of non-volatiles were performed using R on the basis of their signal abundances in hot and cold coffee.

## 3. Results and Discussion

### 3.1. Volatile Compound Profile

As shown in Table 1, a total of 40 volatile compounds were identified, including 12 pyrazines, 9 phenols, 7 furans, 4 aldehydes, 3 ketones, and others. To further evaluate the potency of aroma compounds to the flavor profile, OAV was determined, as shown in Table 1. As shown in Figure 1, concentrations of 13 volatiles in coffee exceeded their respective threshold value, and thus these compounds with OAV > 1 have high chances of being the major aroma contributors to coffee. Among them, 11 aroma-active compounds have been identified in previous research on coffee brewed with French press, coffee machine, and in ready-to-drink coffee [23,24]. 

Pyrazines and furans were reported as the largest shares of volatiles in coffee [25], mainly formed by the Maillard reaction during coffee bean roasting [3]. As key contributors to coffee aroma, OAVs of total pyrazines accounted for over 60% of 13 aroma-active compounds, whose contribution in cold brew coffee was almost as twice as that of in hot brew coffee (Table 1), among which, 2-methylpyrazine was only detected in cold brew coffee, requiring further study to be clarified. Although abundant in coffee, furans made limited contributions to coffee aroma, due to relatively high recognition thresholds of furans [3]. Only the furfuryl acetate was identified as aroma-active furan (Figure 1), with higher OAV of 4.23 in cold brew coffee than in hot brew coffee. Generally, the high temperature increased the saturated vapor pressure, thereby adding the losses of volatile compounds [26]. It can be used to explain higher contents of pyrazine and furan compounds in cold brew coffee, which was consistent with previous studies [9]. Moreover, 4-ethylguaiacol, 3-ethylphenol, 4-vinylguaiacol, and guaiacol, as common phenolic volatiles identified as aroma-active compounds of coffee, were produced from the degradation of chlorogenic acids during green coffee bean roasting [3]. Different phenolic volatile change tends were observed between hot and cold brew coffee, which could be explained in that brewing methods could affect concentrations of these compounds through both the extraction temperature and time.

Among aroma-active compounds, linalool and methyl salicylate were only detected in cold brew coffee, and the content of N-methylpyrrole-2-carboxaldehyde in cold brew coffee was significantly higher than hot brew (Table 1). During the fermentation process of green coffee beans, linalool and methyl salicylate can be produced by various microorganisms [27], which can still keep in detectable concentrations after coffee bean roasting [28]. As an endogenous component in coffee, however, N-methylpyrrole-2-carboxaldehyde was potentially hazardous, which needs to be avoided during coffee preparation [29].

### 3.2. Differential Non-Volatiles Identified in Hot and Cold Brew Coffee 

Sugars and organic acids were basic taste compounds in coffee. As shown in Table 2, sucrose had the largest share of the sugar content, with 15.37 ± 0.86 μg/mL and 14.26 ± 0.72 μg/mL in hot and cold brew coffee, respectively. Consistent with other research, sucrose accounted for the majority of the coffee sweet non-volatiles in coffee beans [41]. Chen and Ho [42] reported that monosaccharides were involved in the Maillard reaction for flavor generation during coffee bean roasting, which might lead to much lower contents of fructose and glucose in coffee. As for the organic acid profile in coffee, no significant difference was observed in most organic acids (Table 2). In a word, a few differences were detected in the sugar and organic acid profiles between hot and cold brew coffee. Moreover, Casas et al. [43] reported that levels of overall amino acids dropped to about 1% of green coffee beans after roasting, which made limited contribution to non-volatile profile of coffee.

Therefore, UPLC-MS/MS was applied to clarify the difference of phenolic acids, alkaloids, and flavonoids between hot and cold brew coffee. Our results showed a total of 322 non-volatiles were identified in hot and cold brew coffee. As shown in Appendix A, hot and cold brew coffee could be distinguished. Moreover, the mix samples (quality controls, QC) were clustered in the center of the PCA score plot, which indicated that the instrument exhibited high stability during data acquisition. Differential non-volatiles were identified by a variable importance in projection score of ≥ 1 and a fold change score of ≥ 2 or ≤ 0.5, which are visually shown in Figure 2A and Table 3. Hot and cold brew coffee were discriminated by 18 differential non-volatiles, including 8 phenolic acids, 5 terpenoids, 3 alkaloids, 1 flavonoid, and 1 coumarin. As shown in Figure 2B, 16 differential non-volatiles showed higher abundance in hot brew coffee, which could be explained in that higher brewing temperature flavored the kinetic energy of water molecules. Due to increased mobility and higher physical forces, non-volatiles were leached out more efficiently from the coffee bed [8].

As essential bioactive compounds in coffee [3], the abundance of 3-caffeoylquinic acid, 4-caffeoylquinic acid, 5-caffeoylquinic acid, caffeine, and trigonelline was shown to be comparable in hot and cold brew coffee (Figure 2A), which was in agreement with previous studies [13,16]. Cordoba et al. [4] reported that over 80% of soluble materials were found to be extracted after 5 min of hot filter brewing. As for cold brew coffee, concentrations of 3-caffeoylquinic acid and caffeine were described to be stable after 400 min [13]. Therefore, it may indicate that in our study, with either high-temperature-short-time or low-temperature-long-time brewing methods, the extraction of caffeine, trigonelline, and caffeoylquinic acids in hot and cold brew coffee might have reached equilibrium due to relatively high solubility in water. 

### 3.3. Color Analysis

The L*a*b* values of hot brew coffee were L* = 29.76 ± 1.70, a* = 23.21 ± 0.61, and b* = 38.57 ± 2.43, while the color of cold brew coffee was L* = 38.79 ± 0.78, a* = 19.14 ± 0.91, and b* = 50.19 ± 3.95. The total color difference value ΔE = 15.27 suggested that the visual appearance of hot and cold brew coffee prepared from the identical coffee beans could be easily distinguished, which was consistent with our sensory quantitative descriptive study (Table 4). To be more specific, hot brew coffee presented darker lightness and redder tone, while cold brew coffee appeared lighter and yellower, which was also observed in medium-roasted coffee brews originating from El Salvador [45]. However, different trends in color changes may be shown in brews from different coffee bean origins, despite similar roasting levels.

The brown color of coffee is related to melanoidins, a class of high-molecular-weight products originating from the Maillard reaction [46]. Melanoidin compounds are more available for extraction with higher temperatures, likely due to the increase in the temperature-dependent solubility [12]. Therefore, hot brewing might make it easier for melanoidin extraction, which contributes to the darker color.

### 3.4. Sensory Evaluation and Differential Flavor Markers 

In the sensory triangle test, a total of 25 panelists correctly identified the coffee sample that was different from others, which was greater than the critical value of 23 (α = 0.05), indicating the significant sensory difference between hot and cold brew coffee could be perceived. To further clarify differential sensory attributes between hot and cold brew coffee, results of the coffee sensory QDA are shown in Table 4. 

Hot brew coffee appeared darker and tasted bitterer and more astringent over cold brew coffee (Table 4), consistent with prevoius studies [11]. Phenolic acids, alkaloids, and flavonoids were reported as main contributors to the bitterness and astringency of coffee [3]. Dicaffeoylshikimic acid, 2-methoxycinnamic acid, and thihydroxycinnamoylquinic acid, as identified phenolic acids in this study (Figure 2B), might be used to explain the difference of bitter and astringent attributes between hot and cold brew coffee. Moreover, as one of the flavonol-3-glycosides, quercetin-3-O-(6″-O-p-coumaroyl) galactoside was identified to be the differential non-volatile compound with a higher abundance found in hot brew coffee. Flavonol-3-glycosides were reported to induce a velvety and mouth-coating sensation with low thresholds, which contributed to astringency in tea and red wine [47,48], although without bitterness on their own, flavonol-3-glycosides were found to amply the bitter taste of caffeine in tea [47]. Therefore, further study is needed to confirm the contribution of quercetin-3-O-(6″-O-p-coumaroyl) galactoside to the bitter and astringent tastes in coffee (Figure 2B). In addition to non-volatile compounds, 4-ethylphenol and 4-ethylguaiacol were reported to increase the intensity and persistence of astringency of the flavanol solutions in the wine system [49]. Therefore, volatile contents might also need to be considered when concerning the bitter and astringent perception of coffee (Table 1). 

As shown in Table 4, cold brew coffee showed higher fruitiness and sweetness intensity. Coniferaldehyde, as an identified key non-volatile in hot brew coffee, was reported to attenuate the fruity perception in red wine, even with concentrations lower than the threshold [50]. It implied that the fruity sensation in hot brew coffee might be decreased by the presence of coniferaldehy (Figure 2B). Sweet and fruity perception of coffee was not only determined by non-volatile but also volatile compounds in foods. Linalool as an aroma-active compound with sweet and fruity notes [51], showing a higher level in cold brew coffee compared with hot brew coffee (Table 1). Interestingly, Barba et al. [52] have reported that linalool in fruit could significantly improve the sweet flavor perception of fruit juice. Hence, a similar result could be found in the presence of furfuryl acetate with the sweet smell in coffee [24]. Moreover, Yu et al. [53] reported that benzaldehyde and furfural increased the intensity of sweet aroma perception in *Huangjiu* even with OAVs < 1. Thus, higher contents of benzaldehyde, furfural, linalool, and furfuryl acetate might also enhance the sweet and fruity perception in cold brew coffee (Table 1). However, contributions of these differential flavor compounds would need further verification on the basis of more extensive investigations on larger scale of hot and cold brew coffee production, involving more general coffee consumers.

## 4. Conclusions

In conclusion, the effects of hot and cold brewing on the flavor profile of coffee appeared significantly different, with 13 aroma-active compounds and 18 differential non-volatiles identified using chromatography–mass spectrometry and multivariate statistical analysis. Hot brew coffee was more closely associated with differential non-volatiles, while cold brew coffee was more associated with aroma-active compounds. A higher abundance of quercetin-3-O-(6″-O-p-coumaroyl) galactoside might be correlated with stronger bitterness and astringency in hot brew coffee, whereas higher OAVs of aroma-active furfural acetate and linalool led to the fruitier and sweeter flavor in cold brew coffee. These findings provide an insightful understanding of the coffee flavor control based on the sensory perception due to brewing methods during coffee industrial processing.

## Figures and Tables

**Figure 1 foods-11-02968-f001:**
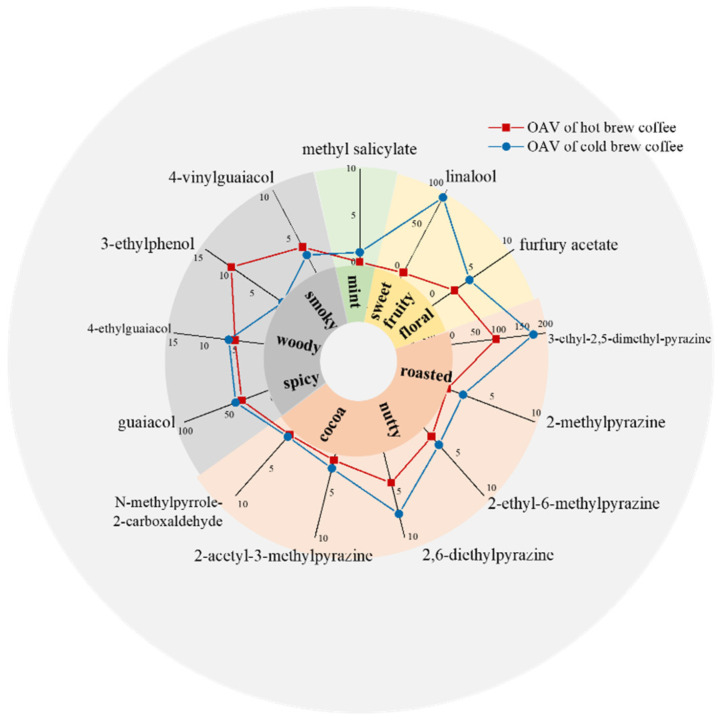
Aroma-active compounds identified using OAV > 1 in hot and cold brew coffee.

**Figure 2 foods-11-02968-f002:**
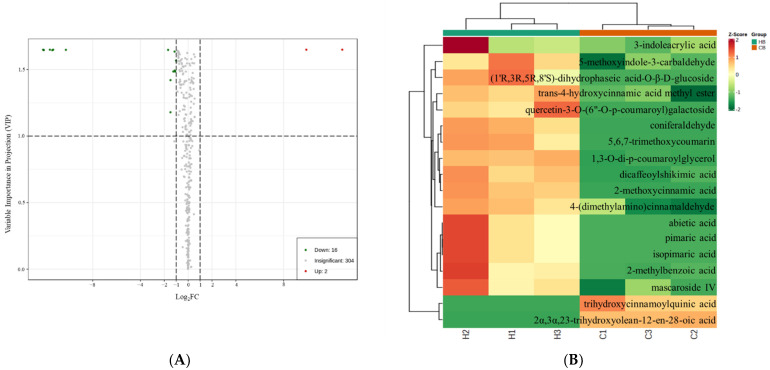
Volcano plot (**A**) and hierarchical cluster analysis (**B**) of differential non-volatiles identified between hot and cold brew coffee. (**A**) Red dots represent differential compounds of which the abundance was upregulated; green dots represent differential compounds of which the abundance was downregulated; and gray dots represent compounds with insignificant differences. If the abscissas had large absolute values, the FC values were also observed to be large. (**B**) Red represents a relatively high content of the compound, whereas green represents a relatively low content. HB: hot brew coffee; CB: cold brew coffee.

**Table 1 foods-11-02968-t001:** Volatile compounds identified in hot and cold brew coffee using head space solid phase microextraction coupled with gas chromatography–mass spectrometry (HS-SPME-GC-MS).

Volatile Compounds	CAS	Odor ^1^	Threshold (μg/L) ^2^	Concentration (μg/L) ^3^		OAV ^4^	
				Hot Brew Coffee	Cold Brew Coffee	Hot Brew Coffee	Cold Brew Coffee
Pyrazine							
2-Methylpyrazine	109-08-0	nutty, cocoa, roasted [30]	60 [31]	n.d.	107.07 ± 4.88		1.78
2,5-Dimethyl pyrazine	123-32-0	cocoa, roasted, nuts [30]	2600 [31]	32.06 ± 1.58	45.45 ± 3.15 *		
2,6-Dimethyl pyrazine	108-50-9	cocoa, roasted, nuts [30]	3100 [31]	33.95 ± 1.45	49.45 ± 3.35 *		
Ethylpyrazine	13925-00-3	peanut, butter, musty [30]	6000 [31]	31.96 ± 2.12	48.64 ± 3.38 *		
2-Ethyl-6-methylpyrazine	13925-03-6	roasted, potato, roasted [30]	30 [31]	47.17 ± 1.59	81.76 ± 4.24 *	1.57	2.73
2-Ethyl-5-methylpyrazine	13360-64-0	coffee, bean, nutty [30]	100 [31]	41.07 ± 1.55	68.02 ± 3.73 *		
2,6-Diethylpyrazine	13067-27-1	nutty, hazelnut [30]	6 [32]	23.94 ± 0.99	44.47 ± 2.39 *	3.99	7.41
3-Ethyl-2,5-dimethyl-pyrazine	13360-65-1	potato, cocoa, roasted [30]	1 [31]	92.91 ± 4.89	173.09 ± 9.21 *	92.91	173.09
2-Ethenyl-6-methylpyrazine	13925-09-2	hazelnut [30]	-	13.79 ± 1.05	22.22 ± 1.01 *		
3,5-Diethyl-2-methyl-pyrazine	18138-05-1	nutty, meaty, vegetable [30]	-	n.d.	82.51 ± 7.18		
2-Methyl-5-[(E)-1-propenyl]pyrazine	18217-82-8	sweet, earthy [33]	-	n.d.	11.97 ± 0.61		
2-Acetyl-3-methylpyrazine	23787-80-6	nutty, flesh, roasted [30]	20 [31]	30.15 ± 0.83	48.10 ± 4.45 *	1.51	2.40
Phenol							
Guaiacol	90-05-1	phenolic, smoke, spice [24]	1.6 [31]	55.37 ± 1.52	66.34 ± 5.90	34.61	41.46
Phenol	108-95-2	phenolic, plastic rubber [30]	2400 [23]	49.29 ± 4.78	44.17 ± 2.59		
4-Ethylguaiacol	2785-89-9	spicy, smoky, bacon [24]	16 [34]	81.78 ± 4.36	97.56 ± 3.95 *	5.11	6.10
3,4-Dimethylphenol	95-65-8	ink, hay [30]	1200 [32]	8.57 ± 0.76	n.d.		
m-cresol	108-39-4	medicinal, woody, leather [30]	31 [32]	14.97 ± 1.39	18.20 ± 1.03		
1-Hydroxy-2,3-dimethylbenzene	526-75-0	phenolic, chemical, musty [30]	500 [32]	4.86 ± 0.20	n.d.		
3-Ethylphenol	620-17-7	leather, ink [34]	1.7 [32]	16.97 ± 0.60	n.d.	9.98	
4-Vinylguaiacol	7786-61-0	dry woody, clove, amber [24]	19 [34]	59.02 ± 4.04*	40.63 ± 2.14	3.11	2.14
2,4-di-tert-Butylphenol	96-76-4	phenol [34]	500 [32]	27.17 ± 5.23	14.31 ± 3.52		
Furan							
Furfural	98-01-1	sweet, woody, almond [30]	3000 [31]	134.23 ± 1.55	200.20 ± 13.44 *		
2-Acetylfuran	1192-62-7	sweet, balsam, almond [30]	10000 [31]	43.33 ± 2.30	54.65 ± 5.03 *		
Furfuryl acetate	623-17-6	sweet, fruity, banana [30]	100 [31]	228.24 ± 5.61	422.76 ± 19.97 *	2.28	4.23
5-Methyl furfural	620-02-0	spice, caramel, maple [30]	6000 [3]	204.70 ± 3.26	268.20 ± 18.65 *		
2-Furanmethanol	98-00-0	sweet, creamy, vanilla [35]	2000 [32]	242.91 ± 6.06	317.66 ± 17.00 *		
1-Furfurylpyrrole	1438-94-4	plastic, green, waxy [30]	100 [32]	70.77 ± 1.11	83.27 ± 10.07		
2,3-Dihydrobenzofuran	496-16-2	floral [36]	-	5.05 ± 0.46	n.d.		
Aldehyde							
Benzaldehyde	100-52-7	sweet, bitter, almond [30]	350 [32]	29.01 ± 1.49	44.22 ± 3.53 *		
N-Methylpyrrole-2-carboxaldehyde	1192-58-1	roasted, nutty [30]	37 [32]	46.82 ± 0.32	58.00 ± 4.11 *	1.27	1.57
2-Phenyl-2-butenal	4411-89-6	sweet, narcissus, cortex [30]	883.8 [37]	8.60 ± 0.34	11.07 ± 2.28		
1H-Pyrrole-2-carboxaldehyde	1003-29-8	musty, beefy, coffee [30]	65000 [32]	19.72 ± 0.77	25.84 ± 2.07 *		
Ketone							
Maltol	118-71-8	sweet, caramel, cotton candy [38]	5800 [23]	29.79 ± 1.10	43.62 ± 0.84 *		
3-Ethyl-2-hydroxy-2-cyclopenten-1-one	21835-01-8	sweet, caramel, maple [30]	53.35 [39]	10.99 ± 1.07	15.92 ± 1.27 *		
4-Hydroxy-3-methylacetophenone	876-02-8	meidical, smoky [40]	-	10.37 ± 1.15	15.48 ± 0.45 *		
Others							
Methyl salicylate	119-36-8	wintergreen, mint [30]	40 [32]	n.d.	41.64 ± 9.20		1.04
3,4-Dimethoxystyrene	6380-23-0	green, floral, weedy [30]	-	12.07 ± 0.59	14.35 ± 0.57 *		
1-(1H-pyrrol-2-yl)-Ethanone	1072-83-9	musty, nut skin, maraschino [30]	170000 [32]	44.15 ± 2.54	56.17 ± 8.66		
1-Acetyl-1,4-dihydropyridine	67402-83-9	-	-	10.85 ± 0.16	n.d.		
Indole	120-72-9	animal, floral, moth ball [30]	40 [32]	5.20 ± 0.07	n.d.		

^1^ Odor description found in the literature; minus sign (-) indicates the odor description was not available in the literature. ^2^ Odor thresholds in water taken from the literature; minus sign (-) indicates the threshold was not available in the literature. ^3^ Values are given as means ± standard deviation (n = 3 independent measurements). * in the same row means significant difference at *p* < 0.05. n.d. means not detected due to the concentration of the given compound being below the detection limit. ^4^ OAV, odor activity value. Only OAV values greater than 1 are presented.

**Table 2 foods-11-02968-t002:** Sugar and organic acid compounds identified in hot and cold brew coffee.

Non-Volatile Compound	Hot Brew Coffee (μg/mL)	Cold Brew Coffee (μg/mL)
Sugar		
Sucrose	15.37 ± 0.86 *	14.26 ± 0.72
Maltose	2.73 ± 0.43 *	2.30 ± 0.14
Galactose	2.44 ± 0.23	2.68 ± 0.10 *
Arabinose	1.87 ± 0.10	2.14 ± 0.10 *
Glucose	0.87 ± 0.17	1.22 ± 0.09 *
Fructose	0.60 ± 0.21	0.61 ± 0.08
Total	23.29 ± 1.32	22.60 ± 0.93
Organic acid		
Quinic acid	696.82 ± 7.84	696.19 ± 12.52
Shikimic acid	33.19 ± 3.62	28.57 ± 0.43
Malic acid	145.17 ± 7.33	160.37 ± 7.09
Tartaric acid	1.14 ± 0.07	1.21 ± 0.02
Oxalic acid	53.68 ± 0.40	54.36 ± 1.16
Citric acid	347.05 ± 15.60	382.12 ± 5.17 *
Total	1277.06 ± 22.21	1322.81 ± 22.07

Values are given as means ± standard deviation (n = 3 independent measurements). * in the same row means significant difference at *p* < 0.05.

**Table 3 foods-11-02968-t003:** Differential non-volatiles identified in hot and cold brew coffee using ultra-performance liquid chromatography–tandem mass spectrometry (UPLC-MS/MS) with the VIP ^1^ ≥ 1 and the fold change score of ≥ 2 or ≤ 0.5.

**Differential Compounds**	**CAS ^2^**	**Formula**	**Molecular Weight (Da)**	**Ionization Model**	**Precursor Ions (Da)**	**Product Ions (Da)**	**VIP**	**Fold** **Change**	**Identification in References ^3^**
1,3-O-di-p-Coumaroyl glycerol	-	C_21_H_20_O_7_	384.12	[M-H]^-^	383.12	163.04	1.65	0.31	-
trans-4-Hydroxycinnamic acid methyl ester	19367-38-5	C_10_H_10_O_3_	178.06	[M-H]^-^	177.00	145.00	1.49	0.46	-
Dicaffeoylshikimic acid	-	C_25_H_22_O_11_	498.12	[M + H]^+^	499.13	163.04	1.63	0.45	[41]
Pimaric acid	127-27-5	C_20_H_30_O_2_	302.22	[M-H]^−^	301.22	301.22	1.65	0.00023	-
Isopimaric acid	5835-26-7	C_20_H_30_O_2_	302.22	[M-H]^−^	301.22	301.22	1.65	0.00023	-
Abietic acid	514-10-3	C_20_H_30_O_2_	302.22	[M-H]^−^	301.22	301.22	1.65	0.00023	-
Quercetin-3-O-(6″-O-p-coumaroyl) galactoside	-	C_30_H_26_O_14_	610.13	[M + H]^+^	611.14	147.04	1.65	0.00039	-
4-(Dimethylamino)cinnamaldehyde	6203-18-5	C_11_H_13_NO	175.10	[M + H]^+^	176.11	146.10	1.42	0.36	-
5,6,7-Trimethoxycoumarin	55085-47-7	C_12_H_12_O_5_	236.07	[M + H]^+^	237.07	176.04	1.65	0.00082	-
5-Methoxyindole-3-carbaldehyde	10601-19-1	C_10_H_9_NO_2_	175.06	[M + H]^+^	176.10	91.20	1.48	0.42	-
Mascaroside IV	2214215-15-1	C_37_H_46_O_14_	714.29	[M + H]^+^	715.29	207.07	1.48	0.47	[42]
2-Methoxycinnamic acid	6099-03-2	C_10_H_10_O_3_	178.06	[M-H]^−^	177.06	133.07	1.65	0.00032	[43]
Coniferaldehyde	20649-42-7	C_10_H_10_O_3_	178.06	[M + H]^+^	179.07	91.00	1.65	0.00022	[44]
2-Methylbenzoic acid	118-90-1	C_8_H_8_O_2_	136.05	[M-H]^−^	135.05	91.05	1.57	0.50	-
3-Indoleacrylic acid	1204-06-4	C_11_H_9_NO_2_	187.06	[M + H]^+^	188.07	118.07	1.18	0.36	-
2α,3α,23-Trihydroxyolean-12-en-28-oic acid	-	C_30_H_48_O_5_	488.35	[M-H]^-^	487.34	487.34	1.65	7724.93	-
(1’R,3R,5R,8’S)-Dihydrophaseic acid-O-β-D-glucoside	-	C_21_H_32_O_10_	444.20	[M-H]^−^	443.19	161.04	1.65	0.00037	-
Trihydroxycinnamoylquinic acid	-	C_16_H_20_O_10_	372.11	[M-H]^−^	371.10	249.06	1.65	958.27	-

^1^ VIP, variable importance in projection. ^2^ Minus sign (-) indicates the CAS was not available in the literature. ^3^ Minus sign (-) indicates the compound was not reported in coffee before in the literature.

**Table 4 foods-11-02968-t004:** Sensory quantitative descriptive analysis for hot and cold brew coffee.

**Sensory Attribute**	**Hot Brew Coffee**	**Cold Brew Coffee**
Color	4.29 ± 0.76 *	3.29 ± 0.49
Nutty	2.86 ± 1.07	2.29 ± 0.49
Coffee	3.14 ± 0.69	2.86 ± 0.90
Fruity	2.00 ± 1.15	3.14 ± 0.69 *
Sweet	0.71 ± 0.49	1.71 ± 0.95 *
Sour	2.86 ± 0.90	3.29 ± 0.95
Bitter	4.14 ± 0.69 *	2.43 ± 0.53
Astringent	2.86 ± 0.90 *	2.00 ± 0.82

Sensory intensity values are given as means ± standard deviation (n = 7 sensory panelists). * in the same row means significant difference at *p* < 0.05.

## Data Availability

Data is contained within the article or Appendix A.

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
