# Peer review of "Comparative Profiling of Hot and Cold Brew Coffee Flavor Using Chromatographic and Sensory Approaches"

_foods, 2022, doi:10.3390/foods11192968_

Round 1
Reviewer 1 Report
The authors investigated the effect of hot and cold brewing on the volatile and non-volatile chemical components of coffee. They also carried out sensory evaluation of both type of coffees. From their results, 13 aromatic and 18 nonaromatic compounds present differentially in both type of coffees were identified.
This approach is interesting and provides insights into the chemistry underlying the taste and properties of hot and cold brew coffee; however, there are many similar work previously reported. The authors need to state the novelty of their work. Also, they need to discuss how their findings actually can bridge the gap in the literature of this subject matter.
Methodology:
Section 2.2: how was the sampling of coffee beans done? did the authors perform independent analyses? how many replicates were included?
Line 192: 48 undergraduates but 18 males - there are 30 female candidates? Please clarify
Line 199: the number of participants seems low? why only the number of males was mentioned?
Line 260: n represents technical replicates or independent experiments?
Table 3: provide explanation on how the compounds were identified
Author Response
The responses to the editor and reviewers are listed as follows.
Comments from the editors and reviewers:
Reviewer 1
The authors investigated the effect of hot and cold brewing on the volatile and non-volatile chemical components of coffee. They also carried out sensory evaluation of both type of coffees. From their results, 13 aromatic and 18 nonaromatic compounds present differentially in both type of coffees were identified.
Point 1: This approach is interesting and provides insights into the chemistry underlying the taste and properties of hot and cold brew coffee; however, there are many similar work previously reported. The authors need to state the novelty of their work. Also, they need to discuss how their findings actually can bridge the gap in the literature of this subject matter.
Response 1: Thanks for your recognition and invaluable suggestions on the manuscript. As required, we have supplemented some relevant reports and discussed their potential limits to highlight the novelty of our study in line 61-66. Moreover, we have supplemented how our study can fill the gap in the Introduction and Conclusions, which can be seen respectively in line 73-74 and line 388-389.
Point 2: Section 2.2: how was the sampling of coffee beans done? did the authors perform independent analyses? how many replicates were included?
Response 2: Sorry for the unclear statement. We have added more detailed information on the sampling of coffee beans in line 78-81.
Yes, independent analyses were performed to ensure sufficient coffee extraction for cold brewing. The total dissolved solids and total phenolics contents became relatively equilibrium after 6 hrs (Figure A). When diluted cold brew coffee to 2% total dissolved solids, the overall sensory qulalites test suggested the one with 8 hrs had the best flavor balance and the least off-flavor (Figure B), evaluated by 4 experts working in coffee flavor for at least 2 years.
Figure A. Total dissolved solids and total phenolics contents of cold brew coffee with different brewing time (n=3 independent measurements).
Figure B. Overall qualities of cold brew coffee with different brewing time (n=4 sensory panelists). The evaluation score was a 10-point scale from 1 (not perceivable) to 10 (strongly perceivable).
According to the reviewer’s kindly suggestion, we have clarified three replicates in line 101-102.
Point 3: Line 192: 48 undergraduates but 18 males - there are 30 female candidates? Please clarify.
Response 3: Sorry for the unclear statement. We have carefully revised the expression of candidates information in sensory evaluation in line 194.
Point 4: Line 199: the number of participants seems low? why only the number of males was mentioned?
Response 4: Thank you for your valuable suggestion. We totally agreed additional paticipants would bring more informative results in the sensory quantitative descriptive analysis. In fact, we started with over 30 panelists, but only 7 passed all tests after 3 rounds of training. We have supplemented more details on the panel selection proceure in line 201-204. Similar scale of sensory panel (6-8 trained paticipants) was also reported in flavor studies on black tea, cheese and chocolates [1-3].
Sorry for the unclear statement. We have added the female panel number in line 202.
- Huang, A.; Jiang, Z.; Tao, M.; Wen, M.; Xiao, Z.; Zhang, L.; Zha, M.; Chen, J.; Liu, Z.; Zhang, L. Targeted and Nontargeted Metabolomics Analysis for Determining the Effect of Storage Time on the Metabolites and Taste Quality of Keemun Black Tea. Food Chem. 2021, 359, 129950, doi:10.1016/j.foodchem.2021.129950.
- Xiang, Q.; Xia, Y.; Song, J.; Saqib, N.; Zhong, F. Characterization of the Key Nonvolatile Metabolites in Cheddar Cheese by Partial Least Squares Regression (PLSR), Reconstitution, and Omission. Food Chem. 2022, 134034, doi:10.1016/j.foodchem.2022.134034.
- Schlüter, A.; Hühn, T.; Kneubühl, M.; Chatelain, K.; Rohn, S.; Chetschik, I. Comparison of the Aroma Composition and Sensory Properties of Dark Chocolates Made with Moist Incubated and Fermented Cocoa Beans. J. Agric. Food Chem. 2022, 70, 4057–4065, doi:10.1021/acs.jafc.1c08238.
Point 5: Line 260: n represents technical replicates or independent experiments?
Response 5: n represented independent experients, in line 264. According to the reviewer’s kindly suggestion, we have clarified meanings of n in line 264, line 281 and line 377.
Point 6: Table 3: provide explanation on how the compounds were identified.
Response 6: Thank you for your constructive suggestion. We have supplemented methods of identifying differential non-volatile compounds in line 315.
Reviewer 2 Report
Dear authors, the study is intresting, but in this form I can agree with the major parts of this manuscript. Bellow I am sending my personal recomendations to increase quality of study.
Line 15 – metabolomics – the authors did not make metabolomic data analysis in this study, please, rewrite this part.
Line 44 – it’s depending on the coffee variety, please rewrite this part.
Line 53 to 65 – please, rewrite and remove the term of metabolomic analysis, you are not working with metabolomic approach, it is irresponsible to do this on this paper. For the study to be considered a metabolome, you need to improve the analyzes presented a lot, a lot!
Line 75 - What is the origin of raw coffee beans? Why did the authors use already processed roasted coffee? This is the big mistake of the study. It is impossible to make such inferences if you do not have traceability control.
Line 84 - The volume of coffee prepared is low, the authors misestimated the processes of origin of the samples, preparation, and extraction of the coffees.
Line 162 – The sensorial panel is limited to compare the approach.
Line 273 and 274 – Is not relevant to coffee science community, this a old, very old information.
Author Response
The responses to the editor and reviewers are listed as follows.
Comments from the editors and reviewers:
Reviewer 2
Dear authors, the study is intresting, but in this form I can agree with the major parts of this manuscript. Bellow I am sending my personal recomendations to increase quality of study.
Point 1: Line 15 – metabolomics – the authors did not make metabolomic data analysis in this study, please, rewrite this part.
Response 1: Thank you for your valuable suggestion. We have revised this part as follows:
In this study, the effects of hot and cold brewing on the flavor profile of coffee were comprehensively investigated based on chromatographic and sensory approaches.
Point 2: Line 44 – it’s depending on the coffee variety, please rewrite this part.
Response 2: Sorry for the unclear statement. We have revised this part as follows:
In a comparative study on the volatiles in cold and hot brew Arabica coffee from Colombia with the medium roasting degree , it was reported that cold brew coffee with immersion extraction had a higher abundance in total furans and pyrazines than hot brew coffee with French Press brewing.
Point 3: Line 53 to 65 – please, rewrite and remove the term of metabolomic analysis, you are not working with metabolomic approach, it is irresponsible to do this on this paper. For the study to be considered a metabolome, you need to improve the analyzes presented a lot, a lot!
Response 3: We appreciate this very constructive comment. According to the reviewer’s kindly suggestion, we have removed the term metabolomic analysis through out the manuscript in line 15-16, line 47-49, line 152-153, line 180, line 283 and line 382-383.
Point 4: Line 75 -What is the origin of raw coffee beans? Why did the authors use already processed roasted coffee? This is the big mistake of the study. It is impossible to make such inferences if you do not have traceability control.
Response 4: Sorry for the unclear statement. We have supplemented origin of raw coffee beans informatiom in line 78-81.
The authors deeply appreciate this very constructive comment. Already processed and roasted coffee beans were used because the objective of this study was to investigate the effect of coffee brewing method alone. We totally agree that green coffee bean processing and roasting methods would also significantly affect the coffee flavor profile. To focus on the effect of brewing, we started with same batch of commerical beans which guaranteed identical proessing and roasting degree. Traceability control would possibliy be achieved by the manufactor records. To clarify this issue, we have reviesd the Introduction to highlight the our focus on brewing (line 32-39).
Point 5: Line 84 - The volume of coffee prepared is low, the authors misestimated the processes of origin of the samples, preparation, and extraction of the coffees.
Response 5: Thank you for your helpful comments. We agreed the volume of coffee prepared is low as our study were still in laboratory scale, pilot-plant study is in need to provide better understanding of this topic. As suggsted, drawbacks of our study have been added in line 372-375.
The authors appreciate the helpful comments on our coffee preparation. To better clarify this procedure, several studies with similar coffee prepation were supplemented in line 93, line 98 and line 101.
Point 6: Line 162 – The sensorial panel is limited to compare the approach.
Response 6: Thank you for your valuable suggestions. We totally agreed additional panelist would be more informative to compare the approach. In fact, we started with over 30 panelists, but only 7 passed all tests after 3 rounds of training. We have supplemented more detailed informations on the panel selection proceure in line 201-204. Similar scale of sensory panel (6-8 trained paticipants) was also reported in flavor studies on black tea, cheese and chocolates [1-3].
- Huang, A.; Jiang, Z.; Tao, M.; Wen, M.; Xiao, Z.; Zhang, L.; Zha, M.; Chen, J.; Liu, Z.; Zhang, L. Targeted and Nontargeted Metabolomics Analysis for Determining the Effect of Storage Time on the Metabolites and Taste Quality of Keemun Black Tea. Food Chem. 2021, 359, 129950, doi:10.1016/j.foodchem.2021.129950.
- Xiang, Q.; Xia, Y.; Song, J.; Saqib, N.; Zhong, F. Characterization of the Key Nonvolatile Metabolites in Cheddar Cheese by Partial Least Squares Regression (PLSR), Reconstitution, and Omission. Food Chem. 2022, 134034, doi:10.1016/j.foodchem.2022.134034.
- Schlüter, A.; Hühn, T.; Kneubühl, M.; Chatelain, K.; Rohn, S.; Chetschik, I. Comparison of the Aroma Composition and Sensory Properties of Dark Chocolates Made with Moist Incubated and Fermented Cocoa Beans. J. Agric. Food Chem. 2022, 70, 4057–4065, doi:10.1021/acs.jafc.1c08238.
Point 7: Line 273 and 274 – Is not relevant to coffee science community, this a old, very old information.
Response 7: Thank you for the insightful comments. According to the reviewer’s kindly suggestion,
we have removed relevant dicussions about the pH and total titratable acidity of coffee in section 3.2.
Round 2
Reviewer 1 Report
The authors have provided satisfactory responses to my comments. There are no other issues. Thank you.
Author Response
Reviewer 1
The authors have provided satisfactory responses to my comments. There are no other issues. Thank you.
Response: Thanks for your recognition and invaluable comments on the manuscript. According to the reviewer’s kindly suggestion, we have checked and polished our manuscript carefully to make it clear for readers to understand, which were marked up using the “Track Changes” function.
Reviewer 2 Report
Dear authors, thank you for the comments sent and for the changes made to the study.
Author Response
Reviewer 2
Dear authors, thank you for the comments sent and for the changes made to the study.
Response: Thanks for your recognition and constructive suggestions on the manuscript.